# Evaluation of the feasibility and acceptability of an integrative group psychological intervention for people with Multiple Sclerosis: A study protocol

**Evangelia Fragkiadaki**[1]*, **Nikki Cotterill**[2], **Claire Rice**[3], **Jonathan A. Smith**[4], **Isabella E. Nizza**[4]

**1** School of Social Sciences, College of Health, Science and Society, University of the West of England, Bristol, United Kingdom, **2** School of Health and Social Wellbeing, College of Health, Science and Society, University of the West of England, Bristol, United Kingdom, **3** Faculty of Health Sciences, Bristol Medical School, University of Bristol, Bristol, United Kingdom, **4** School of Science, Department of Psychological Sciences, Birkbeck University of London, London, United Kingdom

* Eva.fragkiadaki@uwe.ac.uk

**Data Availability Statement:** No datasets were generated or analysed during the current study.

## Abstract

Multiple Sclerosis (MS) is characterised by significant symptom diversity and complexity. The unpredictability of the symptoms and the emotional and cognitive facets of the disease have a significant impact on the patients' quality of life, relationships and other significant areas of living. Psychological interventions have been found to have moderate effects on quality of life, depression, stress reduction, improvement of wellbeing, anxiety, fatigue, sleep disturbances and emotion regulation. Most interventions so far are based on generic models of therapy which cannot always cover the complexity and unpredictability of MS. The present research project follows from an exploratory mixed method study on the experience of psychological interventions and the impact on the management of MS. The results of that study generated themes that led to the development of an integrative group psychological intervention named MyMS-Ally. The current study aims to explore the feasibility and acceptability of MyMS-Ally intervention and obtain preliminary data on the effects on quality of life, emotion regulation, depression and anxiety through the application of a convergent mixed methods design. People with MS will be recruited at the Bristol and Avon Multiple Sclerosis centre, North Bristol NHS Trust. They will participate in MyMS-Ally group intervention for 8 weeks. Individual semi-structured interviews drawing on Interpretative Phenomenological methodology will be conducted before and after the intervention and at three months follow up. Participants will complete quantitative measures on quality of life, emotion regulation, depression and anxiety before and after the intervention and at one and three months follow up. The aim is to explore the relevance, sustainability and adherence to the intervention and study processes (feasibility) as well as the appropriateness of the intervention based on the emotional and cognitive responses, satisfaction and perceived effectiveness (acceptability). It is aspired that this patient-centred psychological intervention will address needs and preferences of people with MS. The results of the present study will

**Funding:** The current study was funded by the University of the West of England (UWE, Bristol) through the internal funding scheme Vice Chancellor's Early Career Researcher Development Award acquired by EF. The funders had and will not have a role in study design, data collection and analysis, decision to publish, or preparation of the manuscript.

**Competing interests:** The authors have declared that no competing interests exist.

provide data for further development of the intervention and will lead to a big scale evaluation study.

## Introduction

Multiple Sclerosis (MS) is an autoimmune, neurodegenerative condition. People with MS (pwMS) experience a range of physical symptoms such as pain, fatigue, motor weakness, vision impairment, cognitive deficits [1]. There are approximately 106,000 Multiple Sclerosis (MS) cases recorded in the United Kingdom [2]. Living with MS requires continuous behavioural and cognitive modifications for managing physical and psychosocial challenges. Literature has focused extensively on the psychiatric and psychological dimensions of MS [3]. There is a higher prevalence of anxiety and depression for pwMS than the general population [4]. PwMS demonstrate higher emotion dysregulation with significant impact on their quality of life [5]. Quality of life has also been associated with higher self–esteem and resilience [6], developing a sense of coherence and control over MS [7], positive affect, optimism, self–efficacy and meaning making processes [8]. The role of psychological interventions and their positive effects on pwMS' mental health and coping has been well documented [6, 9, 10].

The literature has mainly focused on theory-driven, generic models where Cognitive Behavioural Therapy (CBT) and Mindfulness have been the most prominent ones. Both approaches have been associated with improvement of wellbeing [11, 12], reduction of stress, anxiety, depression, fatigue and sleep disturbances [13–16], cognitive fatigue and depression [17]. Mindfulness interventions have been proven efficient in reducing depressive symptoms and fatigue, increasing the quality of life as well as improving pain management for pwMS [18]. They have also been found to reduce emotion dysregulation for pwMS [19]. More integrative and third–wave approaches such as Acceptance and Commitment Therapy (ACT), Dialectical Behaviour Therapy (DBT) and counselling have been found to contribute to improvement in quality of life, adjustment and coping with MS [10, 20–22]. Group intervention programmes have also been found helpful for pwMS [19, 23]. Studies have also explored different approaches such as relaxation [24], motivational interviewing [25] and hypnotherapy [26] and their effects on stress reduction, improvement of pain and psychological wellbeing have been recorded.

Previous research has mainly been based on quantitative designs that aim to capture the psychological and physical symptom reduction for pwMS as a result of a psychological intervention implementation. Even though the findings show the positive impact of these interventions, little is known about the experience of pwMS that take part in these interventions and especially their needs and preferences with regards to psychological techniques involved in their care. As an example of this, even though quantitative findings showed that telephone-based CBT has positive impact on MS symptom management and psychological processes, qualitative findings were not entirely consistent showing that pwMS value the contact with the therapist and found CBT booklets "overwhelming and overcomplicated" [27, p. 1792].

Generic models of psychological interventions often cannot address the complexity and unpredictability of MS. Based on recent developments of process–based psychological interventions, psychological approaches can be developed based upon the specific characteristics of the condition of pwMS [28, 29]. Psychological approaches can adapt to pwMS' needs and preferences so that efficient development and delivery of psychological techniques can be facilitated [30, 31]. PwMS' voices and accounts can guide psychological intervention programmes as well as methods of evaluation of these programmes. More longitudinal and qualitative

projects are recommended as a way to explore the effects of psychological interventions on the management of MS [31].

The current research project aims to address the above issues and explore the feasibility and acceptability of an integrative group psychological intervention for pwMS named MyMS-Ally through a mixed method design. The development of the intervention was based on pwMS's accounts of what they found helpful and unhelpful in psychological interventions they engaged in as well as the impact on their experience and management of MS [32]. MyMS-Ally techniques were based on psychological interventions that have been found effective for pwMS in previous research conclusions. The development of the intervention also followed consultation with mental health practitioners that facilitate groups with pwMS as well as the Patient and Public Involvement (PPI) focus group that was conducted as part of the design of the present project. The aspiration has been to establish processes that correspond to procedures, needs and characteristics of pwMS [29].

Following the systematic development of the intervention, a pilot feasibility and acceptability study is essential in order to address uncertainties and make appropriate modifications before further evaluation of the intervention [33]. The current study will provide significant information on how the intervention will be appraised by participants that take part, helpful and unhelpful aspects of the intervention based on the participants' experiences and accounts as well as the feasibility of implementation of the intervention. Moreover, adherence to outcome measures and study processes will be assessed which will give valuable information for the design of the consequent evaluation study [33].

## Objectives

The primary objective of the current study is the investigation of the feasibility (relevance, sustainability and adherence) and acceptability (emotional and cognitive responses, satisfaction and perceived effectiveness) of the group psychological intervention MyMS-Ally. As a secondary objective, the researchers will collect primary data to evaluate the effects of the intervention on quality of life, depression, anxiety and emotion regulation. A convergent mixed method design will be implemented, with the qualitative and quantitative data gathering processes occurring in parallel. Qualitative individual interviews will provide insight into the participants' experience of taking part in the intervention, how they perceive the impact on their experience and management of MS and which intervention processes they perceived as helpful or unhelpful. Quantitative outcome measures will record any potential change. The final integration between qualitative and quantitative findings will allow a more complete understanding of the intervention as a whole. The aim of the study is to reach conclusions with regards to the relevance and appraisals of participants taking part in the intervention so that final modifications are made before the intervention is further evaluated. Additionally, the study aims to reach preliminary conclusions that will indicate the impact of the intervention as well as effects that are meaningful in everyday practice with people with MS [34].

## Materials and methods

This is a mixed–method pilot design with the aim to explore feasibility and acceptability of the integrative, group psychological intervention MyMS-Ally. Quantitative and qualitative data will be collected in order to facilitate analysis of pwMS' experience of the intervention as well as analysis of outcomes on quality of life, anxiety, depression and emotion regulation. The mixed-method design offers the framework for statistically significant conclusions to be made but also incorporate participants' meaning–making accounts with an in-depth description of their experience of the intervention. The study adheres to an idiographic paradigm of exploring the impact of the intervention [35, 36] where each participant's variations and changes

over time are investigated through multiple points of data collection. Following the patient–centred regime of the development of the intervention and the evaluation processes, the aim of the study is to identify changes over time within each participant. Quantitative and qualitative methods are deemed "compatible partners" in this research context [37, p. 463]. The study design is outlined in Fig 1.

## Patient involvement statement

Five members of the Research Network of UK MS Society participated in a focus group with the project Chief Investigator (CI; EF) to discuss MyMS-Ally intervention development and implementation as well as the present research design. They were sent a brief overview of the research project and MyMS-Ally intervention outline prior to the focus group. The CI presented the intervention and the study processes briefly in the beginning of the group before moving on to the discussion. The focus group was semi–structured with specific questions developed by the CI which focused on: a. intervention, b. research processes and c. dissemination. The CI also encouraged open discussion on further feedback or input members might have had on the intervention and research plan. The members' feedback has been incorporated in the intervention planning and research design.

## Participants

Participants recruitment will take place in the Bristol and Avon Multiple Sclerosis Centre in Southmead Hospital, Bristol (BrAMS), North Bristol NHS Trust. A group of 5–6 participants will be recruited to take part in the MyMS-Ally group intervention. The group size reflects the minimum number of members of a group psychological intervention for meaningful interactions to unfold and the maximum number which allows enough time and space for each member to use the group effectively [38]. The size of the sample also reflects the recommendations made in the PPI focus group. Should more people with MS express interest in taking part, the resources will be evaluated for more than one group to be offered.

The study will be presented to the multidisciplinary team of professionals at BrAMS who will then disseminate the research information, participant information sheet and consent form to eligible patients. The potential participants who will be interested in participating or finding out more about the project will contact the CI via phone, text or email (according to their preference) and they will have an initial discussion on what the intervention and study entail. This discussion will also serve as the initial screening where the CI will be able to explore how the potential participants fit the inclusion and exclusion criteria:

**Inclusion criteria.**

- Participants must be able to read, write and speak English.

- Participants must be willing and able to give informed consent for participation in the study.

- Participants can be people who identify as men, women, non–binary, aged 18 and above.

- Participants must be diagnosed with Multiple Sclerosis and have received their first diagnosis more than two years ago.

- Participants must be cognitively and physically ready and willing to engage into eight weekly group psychological intervention sessions (discussed in screening meeting with potential participant).

- Participants should have access to the internet and to a device that will allow them to join the intervention and data collection processes through Microsoft Teams programme.

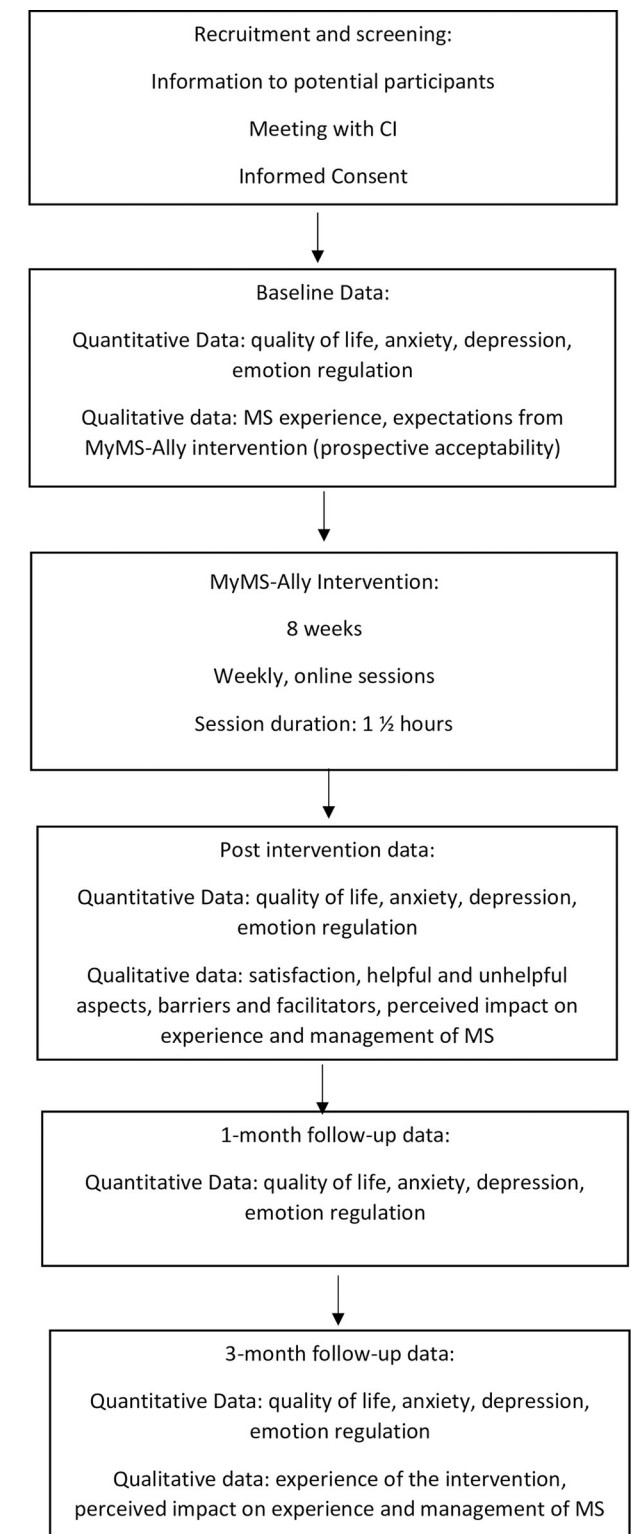

**Fig 1. Outline of study design.**

**Exclusion criteria.** The participants may not enter the study if ANY of the following apply:

- People with MS with suspected or diagnosed comorbidity with Depression, Bipolar Disorder or Psychotic Disorders (discussed at the screening stage with CI).

- People with MS that have received the MS diagnosis in the last two years prior to data collection point (avoid effects of newly diagnosed people with MS who would still be contemplating and making therapy decisions, there are specific needs that characterise this population).

- People with MS who are attending another psychotherapy or psychological intervention process at the time of application of MyMS-Ally group intervention.

The patients who will agree to participate will have to return the signed consent form to the CI before the commencement of collection of data and intervention.

## MyMS-Ally group intervention

The intervention has been developed based on previous research conclusions and consultations with practitioners about what is helpful for pwMS in their experience of psychological interventions. Moreover, following the process—based paradigm, the intervention is based on pwMS' accounts of their experience and management of MS as they related them to their experience of psychological interventions [32]. MyMS-Ally group intervention will last eight weeks and participants will attend weekly sessions that will last 1 ½ hours each (8 sessions). The details of the intervention are outlined in Table 1. Following the PPI group recommendations, the intervention will take place online and the pace and the needs of the participants will be accommodated. As an example, the facilitator will establish breaks during the sessions for the participants to move away from their monitors as well as include summaries of previous sessions in the beginning of each session. The group will be facilitated by an accredited Counselling Psychologist with experience in facilitating integrative group interventions.

## Data collection

Feasibility and acceptability of MyMS-Ally online group intervention will be explored mainly through qualitative interviews. The participants will attend three qualitative semi-structured interviews:

1. At baseline, before commencement of the intervention: the interviews will focus on the experience of their MS and the expectations participants have from MyMS-Ally group intervention (prospective acceptability) [39].

2. At the end of the intervention (at the end of the 8 sessions): the interviews will focus on satisfaction, helpful and unhelpful aspects of the intervention as they experienced them, barriers and facilitators to complete the intervention programme and any impact they perceived the intervention had on their experience and management of their MS [39].

3. At three months follow–up: the interviews will focus on their experience of MyMS-Ally intervention and the impact on their experience and management of their MS.

Interviews are expected to last approximately one hour and they will take place online on a platform with encryption where recordings are possible (Microsoft Teams) and stored on the CI's university (UWE) OneDrive in a password protected folder. The interviews will be conducted by an independent researcher who has experience in qualitative data collection based on Interpretative Phenomenological Analysis (IPA) methodology [40].

**Table 1. Outline of MyMS-Ally online group intervention.**

| Sessions | Content |
| --- | --- |
| Week 1: What do emotions have to do with it? | • Introductions and group ground rules<br>• Journey in the group: journal keeping<br>• The mind-body connection: exploring participants' understandings and experiences<br>• End of session exercise: breathing technique (Mindfulness) |
| Week 2: What is MS? Who is my MS? | • Review of previous session.<br>• Relating to my MS: story of their experience of MS starting with diagnosis exploring the sense of self with MS, incorporating MS in identity, losses<br>• Reflections<br>• End of session exercise: breathing technique (Mindfulness) |
| Week 3: The mindful body with MS | • Review of previous session.<br>• Embodied experiences related to MS: relating to the body with MS<br>• Representations of disability: exploring the meaning of "disability"<br>• End of session exercise: breathing technique (Mindfulness) |
| Week 4: The difficult emotions in the journey | • Review of previous session<br>• Explorations of thoughts and feelings (CBT)<br>• Challenging the thoughts: thoughts diffusion technique (ACT), developing alternative thoughts (CBT)<br>• Focus on feelings and experience in the here and now (Mindfulness) |
| Week 5: "Us and Them" | • Review of previous session<br>• Building and redefining relationships after MS diagnosis.<br>• Exploring communication patterns and interactions: emotion regulation and mentalization techniques<br>• End of session exercise: breathing technique (Mindfulness) |
| Week 6: Every day with MS is a day to celebrate | • Review of previous session.<br>• Living the new life with MS: everyday problem solving and coping mechanisms<br>• Gaining a new pace in life: narratives of adjustment<br>• Reflections and learnings from the sessions: what do we take with us? |
| Week 7: We are not only MS | • Review of previous session.<br>• Reframing self–narrative incorporating my MS<br>• Engaging my MS in discussion: gestalt approach techniques<br>• Reflections<br>• End of session exercise: breathing technique (Mindfulness) |
| Week 8: What I am taking with me | • Review of previous session.<br>• Ending with transferable skills<br>• Reflections on group process<br>• Reflections on group dynamics<br>• End of session exercise: breathing technique (Mindfulness) |

Additional indicators will be recorded in order to explore further the feasibility and acceptability of MyMS-Ally group intervention [39]:

• Number of pwMS referred by the health and mental health professionals of the service.

• Number of pwMS attending the screening meetings with the CI.

• Time taken to complete the questionnaires and missing data.

• Follow up responses (1 month and 3 months follow up).

• Number of sessions attended by the participants.

• Feedback by MyMS-Ally group intervention facilitator.

Participants will complete three questionnaires in order to address the secondary objective of the study. The aim is to collect preliminary quantitative data on outcomes and impact of the intervention on quality of life, anxiety, depression and emotion regulation. Quality of life,

anxiety and depression have been extensively used as indicators for change in the literature of evaluating psychological interventions for people with MS [22]. Emotion regulation is a factor that needs to be explored further as it has been associated with the wellbeing of people with autoimmune diseases [41]. The questionnaires participants will complete are:

- Satisfaction with Life Scale (SWLS): The satisfaction with life scale was developed to assess the participants' satisfaction with life as a whole [42]. It comprises of five items. It has shown sufficient sensitivity to detect change in life satisfaction during the course of a clinical intervention [43] and in the field of health psychology it has been used to evaluate subjective quality of life [44]. The items are global rather than specific in nature. Statements are rated on a 7-point Likert scale (1 strongly disagree, 7 strongly agree). The SWLS has been shown to be a highly reliable, valid, and responsive measure of overall quality of life [43]. Test–retest reliability for the scale has been reported to be .82 [42].

- Hospital Anxiety and Depression Scale (HADS): The HADS is a self–report scale assessing the states of depression and anxiety. Participants are asked to reflect on how they have been feeling during the past week. The scale comprises of 14 items, seven for anxiety and seven for depression. The questionnaire has demonstrated good factor structure and internal consistency, with values of Cronbach's coefficient (a) 0.80 for the anxiety subscale and 0.76 for the depression subscale [45]. The questionnaire has also been used in research with pwMS [46].

- Emotion Regulation Questionnaire (ERQ): A 10-item scale designed to measure respondents' tendency to regulate their emotions in two ways: (1) Cognitive Reappraisal and (2) Expressive Suppression. Respondents answer each item on a 7-point Likert-type scale ranging from 1 (strongly disagree) to 7 (strongly agree). Higher scores in the reappraisal scale indicate better ability to monitor, evaluate, and reduce distress. In contrast, higher scores in the suppression scale indicate poorer ability to repair mood and to manage stressful situations. Overall, the ERQ has very good psychometric properties with a definite two-factor structure, good internal consistency (0.83 and 0.79 for reappraisal and suppression, respectively), and satisfactory test–retest reliability [47]. Cognitive reappraisal has been associated with wellbeing for people with Multiple Sclerosis [48] and has been used to assess psychological states and quality of life for pwMS [49, 50].

A significant advantage of the questionnaires chosen is that they are brief (it takes about 10–15 minutes to complete). Participants will complete the questionnaires four times during the course of the study:

1. At baseline, before commencement of the intervention

2. At the end of the intervention

3. At one month follow up after the end of the intervention

4. At three months follow up after the end of the intervention

## Methods of analysis

### Qualitative data

The collection, analysis and final presentation of findings of the qualitative data will draw on guidelines of Interpretative Phenomenological Analysis [40]. The analysis will be thorough and detailed, and every participant's interviews will be analysed separately on a case-by-case basis. The CI will first listen to the recording of each interview in order to gain familiarity with

each transcript and make first observations, noting linguistic, descriptive, and conceptual codes throughout the transcripts. A longitudinal analysis will be conducted for each case conveying the diachronic developmental dimension in terms of their experience of the intervention and the impact on the experience and management of their MS [51]. A case-by-case analysis and a careful and exhaustive examination across cases will be conducted to reveal the divergences and commonalities amongst the cases. The focus will be on the subjective experience of the MyMS-Ally psychological intervention and how participants appraise, perceive the intervention processes helpful or unhelpful and any change processes they might experience in relation to intervention processes. Analysis will be cyclical and iterative, constantly going back and forth in the data. Personal experiential statements will be grouped in tables and diagrams and discussed in the research team, who will review all the steps of the analysis. Group experiential themes across cases will be developed. Engaging in a more interpretative level of analysis, the meaning of each group experiential theme will be further discussed as grounded on and related to the participants' narratives.

## Quantitative data

Descriptive statistics (means and standard deviations) will be calculated to provide insight into the participants' characteristics as well as indication of any potential changes in the mean scores between the different administrations of the measures at four different points in time (baseline, end of intervention, 1 month and 3 month follow up). Within subjects effects will also be calculated and reported. The final quantitative findings will be presented in a table reporting F, df, effect sizes, CI and p values.

## Ethical considerations and declarations

The study has been approved by National Health Service Health Research Authority and Health Care Research Wales (REC reference: 22/EM/0100). Ethical approval has also been obtained by the Faculty Research Ethics Committee on behalf of the University of the West of England (Reference: HAS.22.06.127).

The participants must personally sign and date the Informed Consent form before any study specific procedures are performed. Written versions of the Participant Information Sheet and Informed Consent will be presented to the participants detailing the exact purpose and nature of the study, what it will involve and the known possible risks and benefits involved in taking part. It is clearly stated that the participants are free to withdraw from the study for any reason without prejudice to future care, without affecting their legal rights, and with no obligation to give the reason for withdrawal.

The participants will have this information long enough before making an informed decision to take part in the study. The maximum timeframe is two weeks after the initial screening meeting with the CI. Participants will also have the opportunity to ask the CI or other independent parties any questions as they decide whether they will participate in the study. A copy of the signed Informed Consent will be given to the participant. The original signed form will be retained in the password protected folder in the CI's university OneDrive (UWE). The health professional who referred the participant to the study will be informed verbally by the CI that their service user is taking part in the study.

The roles of each member of the research team are clearly defined. The CI is the lead researcher of the study and is not involved in the implementation of the group psychological intervention or the collection of the qualitative data. The group facilitator is an accredited Counselling Psychologist who will be supervised by a senior practitioner; they will comprise the clinical team of the study who will not be involved in the collection and analysis of the

data. The qualitative interviews will be conducted by an experienced independent researcher. Therefore, research and clinical processes are clearly separated in the course of the study. The CI's identity as an accredited Counselling Psychologist, her clinical skills and her experience working with pwMS in research will provide the means to create a safe and respectful environment for participants in the processes of intervention implementation and collection of data.

At the end of the study participants will have the opportunity to meet with the CI for check–in meetings should they accept the invitation. The final results of the study will also be fed back to the participants should they accept the report. Data will be stored in the University of the West of England Research Repository and will be reserved there as restricted data, accessed by third parties for purposes of review. The participants' personal details will be destroyed according to guidelines.

The study processes have been reviewed by the supervisor and collaborators of the project and it has been thoroughly discussed in the PPI group where no major amendments were recommended. The analysis of the data will be discussed in the group of researchers and an experienced qualitative researcher external to the project will audit the analysis of the qualitative data to ensure integrity and trustworthiness.

## Data management plan

The minimum necessary person–identifiable information will be collected for the purposes of the study. The study will comply with the General Data Protection Regulation (GDPR) and Data Protection Act 2018, which require data to be de-identified as soon as it is practical to do so. The processing of the personal data of participants will be minimised by making use of a unique participant study pseudonym on all study documents and any electronic database. All documents will be stored securely on the University of the West of England OneDrive in password–protected folders and will only be accessible by study staff. The CI will safeguard the privacy of participants' personal data.

The personal details that will be collected from the participants will include name, surname, email, phone number so that the CI will have a way to contact participants for the intervention implementation and the collection of data arrangements. No other members of the research team will have access to participants' personal details.

## Status and timeline of the study

Ethical approvals have been obtained and the study is now at the participant recruitment stage. The healthcare professionals in BrAMS present the information to eligible patients who then contact the CI for the screening meeting. Baseline data have been collected and the intervention has been implemented and completed in May 2023. Post-intervention and follow up data will be collected by the end of summer 2023.

## Discussion

The positive impact of psychological interventions on the experience and management of MS has been well documented in the literature. NICE guidelines include elements of CBT and Mindfulness as means of self- management for fatigue for pwMS [52]. The current study adds to the discussion following a patient-centred, integrative modality of psychological intervention development. The recent focus on idiographic, process-based methods of intervention development as well as evaluation of interventions [28, 29] provide the framework for MyM-S-Ally group intervention development. The needs and the preferences of pwMS guided the themes of each session of the intervention and the psychological techniques implemented to process these themes. It is aspired that the intervention will enhance self–management of MS

symptoms and provide patients with transferable skills which they can implement following the end of the intervention.

Psychological interventions developed especially to address the complexity of MS and the experience and management of MS symptoms are not broadly discussed in the literature which is dominated by mainstream CBT and Mindfulness approaches. The current study aims to address this gap. The mixed method design will provide the space for participants to provide accounts of their expectations, their experience of the intervention, what they find helpful and unhelpful as well as any impact the intervention might have on their experience and management of MS in their everyday lives. Mixed method designs for feasibility and acceptability studies are also encouraged by the Medical Research Council [33]. The results of the study will provide the information for further development of the intervention and the template for a case-series evaluation on a larger scale.

## Strengths and limitations

The results of the study must be considered with caution given the small sample number. Participants will not be randomised and there will not be a control group. Therefore, there are limitations with regards to controlling confounding factors and producing generalisable findings. However, the idiographic nature of the study, the multiple points of collection of quantitative as well as qualitative data will provide a rich database which will lead to valuable conclusions for the feasibility and acceptability of MyMS-Ally psychological intervention. Moreover, the fact that participants will be recruited from one service does not make the sample representative. However, this is a small-scale feasibility and acceptability study which will offer the necessary conclusions upon which a larger-scale, case-series, evaluation study with more than one group will be designed and implemented. The research group has distinct roles to limit potential conflict and reduce the impact of bias on data collection and analysis processes.

## Dissemination plans

The results of the study will be published in peer-reviewed journals and conference presentations. Moreover, the authors aim to disseminate the results to a wider audience and public events, seminars, podcasts and workshops will be facilitated. The target groups will be charities and organisations that could potentially benefit from their involvement in the dissemination strategy, for example organisations that offer emotional and other support to pwMS. The dissemination strategy will target pwMS, carers, practitioners as well as people with an interest in psychological intervention development for chronic conditions.

## Conclusion

Process–based methods of intervention development and implementation as well as longitudinal qualitative data collection and analysis are aspired to provide the framework for a patient–centred approach to psychological intervention for pwMS and its evaluation. The study aims to provide an in-depth insight on the pwMS' experience of the intervention, what they found helpful or unhelpful as well as any impact it might have on their experience and management of their MS.

## Supporting information

**S1 Protocol. Feasibility and acceptability study MyMSAlly.**
(PDF)

## Author Contributions

**Conceptualization:** Evangelia Fragkiadaki, Nikki Cotterill, Claire Rice.

**Funding acquisition:** Evangelia Fragkiadaki.

**Investigation:** Evangelia Fragkiadaki.

**Methodology:** Evangelia Fragkiadaki, Jonathan A. Smith, Isabella E. Nizza.

**Project administration:** Evangelia Fragkiadaki.

**Resources:** Evangelia Fragkiadaki, Claire Rice.

**Supervision:** Nikki Cotterill.

**Writing – original draft:** Evangelia Fragkiadaki.

**Writing – review & editing:** Evangelia Fragkiadaki, Nikki Cotterill, Claire Rice, Jonathan A. Smith, Isabella E. Nizza.

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
