## [Decision Letter · Decision Letter 0]

23 May 2023

PONE-D-22-34801

Evaluation of the feasibility and acceptability of an integrative group psychological intervention for people with Multiple Sclerosis: a study protocol

PLOS ONE

Dear Dr. Eva Fragkiadaki & co-authors,

Thank you for submitting your manuscript to PLOS ONE. After careful consideration, we feel that it has merit but does not fully meet PLOS ONE’s publication criteria as it currently stands. Therefore, we invite you to submit a revised version of the manuscript that addresses the points raised during the review process.

Your study protocol deserves much admiration for addressing an important area of need in psychology; managing patients with MS. The intervention and the research project seem to be well-planned. In addition to the two reviewers' comments, I wish you to clarify/address the following aspects.

1. Suggest briefing the relevant quantitative and qualitative methodologies adopted in the abstract.

2. Line 427: Status and timeline of the study. - I understand there was a significant delay in peer-reviewing your article. However, please update the current study stage, indicating the actual time as well.

<ul><li> 

A rebuttal letter that responds to each point raised by the academic editor and reviewer(s). You should upload this letter as a separate file labeled 'Response to Reviewers'.<li> 

A marked-up copy of your manuscript that highlights changes made to the original version. You should upload this as a separate file labeled 'Revised Manuscript with Track Changes'.<li> 

We look forward to receiving your revised manuscript.

Kind regards,

GVM Chamath Fernando,

MBBS PgD-FM DipPallMed MCGP MRCGP

Academic Editor

PLOS ONE

“The current study was funded by the University of the West of England (UWE, Bristol) through the internal funding scheme Vice Chancellor’s Early Career Researcher Development Award acquired by EF. The funders had and will not have a role in study design, data collection and analysis, decision to publish, or preparation of the manuscript.”

“The current study was funded by the University of the West of England (UWE, Bristol) through the internal funding scheme Vice Chancellor’s Early Career Researcher Development Award acquired by EF. The funders had and will not have a role in study design, data collection and analysis, decision to publish, or preparation of the manuscript.”

5. The in-house editorial staff feels that your study meets the World Health Organization definition of a clinical trial because it is a prospective study in which participants were assigned to receive the MyMS-Ally intervention to investigate the effects on quality of life, anxiety, depression, emotion regulation.

Please can you provide the following when you resubmit your manuscript:

- SPIRIT checklist, and - SPIRIT schedule of enrolment, interventions, and assessments as the manuscript’s Figure 1

- Protocol as submitted to the IRB for approval

Please upload a copy of your trial study protocol as a supporting information file. By the study protocol, we

mean the complete and detailed plan for the conduct and analysis of the trial that the ethics committee approved before the trial began. Please send this in the original language. Your study protocol will be made available to the editors and reviewers, and will be published as supporting information with your manuscript if accepted for publication. (If you do not agree to this, we will not be able to publish your manuscript).

- Trial registration

PLOS ONE requires that all clinical trials are registered in an appropriate registry (the WHO list of approved registries is at http://www.who.int/ictrp/network/primary/en/index.html and more information on trial registration is at

http://www.icmje.org/about-icmje/faqs/clinical-trials-registration/). Please state the name of the registry and the registration number (e.g. ISRCTN or ClinicalTrials.gov) in the submission data and on the title page of your manuscript

**Comments to the Author**

1. Does the manuscript provide a valid rationale for the proposed study, with clearly identified and justified research questions?

Reviewer #1: Yes

Reviewer #2: Yes

2. Is the protocol technically sound and planned in a manner that will lead to a meaningful outcome and allow testing the stated hypotheses?

Reviewer #1: Partly

Reviewer #2: Yes

3. Is the methodology feasible and described in sufficient detail to allow the work to be replicable?

Reviewer #1: Yes

Reviewer #2: Yes

4. Have the authors described where all data underlying the findings will be made available when the study is complete?

Reviewer #1: No

Reviewer #2: Yes

5. Is the manuscript presented in an intelligible fashion and written in standard English?

Reviewer #1: No

Reviewer #2: Yes

6. Review Comments to the Author

You may also provide optional suggestions and comments to authors that they might find helpful in planning their study.

Reviewer #1: The authors indicate that standard CBT practices have been re-directed to address issues of MS. This is a useful endeavor. However, reading through the little information provided on the content of the intended psychotherapy intervention, I do not see anything that novel the authors have included in their MyMAS-Ally. Hence, new contribution to the literature is minimal, in my opinion. If the authors think otherwise, they could provide more details on HOW their intervention actually offers novel input to the psychological aspects of MS. Further, in the inclusion criteria, the last bullet point, could those diagnosed with MS do this?. And, the objectives stated in article seems to be for the overall study whilst the article per se only reports a section of this overall study – this is confusing to the reader.

Reviewer #2: Very interesting topic and well discussed. The study clearly state, who will be the samples (inclusion and exclusion), how will the intervention be implemented, who will conduct the intervention, the duration of the intervention as well as the data collection for pre, post and follow-up and also the information related to the qualititative study. Just a minor one, would like to suggest adding the reliability information for Satisfaction with Life Scale, Hospital Anxiety and Depression Scale (HADS).

7. PLOS authors have the option to publish the peer review history of their article (what does this mean?). If published, this will include your full peer review and any attached files.

Reviewer #1: No

Reviewer #2: No

<quillbot-extension-portal></quillbot-extension-portal>

---

## [Author Response · Author response to Decision Letter 0]

7 Jun 2023

Thank you for your encouraging words for our research project. We strongly believe there if great benefit in the exploration of psychological interventions focused on the people with MS’ needs and preferences and engage into more idiographic methods of evaluating these interventions.

Thank you for the constructive comments. We have amended the article following the feedback. Please find below each point of the feedback along with our response to it.

Editors’ feedback:

1. Briefing the relevant quantitative and qualitative methodologies adopted in the abstract: more details on the methods have been added in the abstract (p. 4, line 85).

2. Update the timeline of the study: the timeline of the study has been updated with the progress of study so far (p. 22, line 434).

1. Formatting the article as well as naming files: the format of the article and the file names have been adjusted to adhere to the journal guidelines.

2. Funding statement: I have deleted the funding statement from the manuscript, please include the statement in my Funding Statement online form:

“The current study was funded by the University of the West of England (UWE, Bristol) through the internal funding scheme Vice Chancellor’s Early Career Researcher Development Award acquired by EF. The funders did not have and will not have a role in study design, data collection and analysis, decision to publish, or preparation of the manuscript.”

Thank you.

3. Include captions for supporting information: I have updated the manuscript and included the captions for supporting information in the end of the manuscript (p. 24, line 505).

4. References: Reference 32 has changed (line 625) and no other changes in the references have been made.

5. Clinical trials registration: we appreciate how the editorial has a sense that our study could fall under clinical trial definition. However, this is a feasibility and acceptability study of a psychological intervention that follows a case – series mixed method design, not an experimental or randomised controlled trial design. The project has been thoroughly and robustly reviewed by National Health Service Ethics Health Research Authority (governmental authority) committee as well as University of the West of England (Bristol) Faculty Research Ethics committee and they did not identify this project as a clinical trial or requested registration as such. Following the National Institute for Health and Care Research definition of a clinical trial:

“A clinical trial is a research project that compares two or more treatments in patients with a particular condition or at risk of a condition to help generate high quality evidence about which is the more effective treatment or preventative strategy. The treatment being investigated in a clinical trial can be a medicinal product, a procedure, a device or another type of therapeutic intervention.” (https://www.nihr.ac.uk/documents/clinical-trials-guide/20595).

The current study does not meet these criteria. On the contrary, we advocate the need to move away from large scale clinical trials and focus on more idiographic methods of psychological intervention evaluations. 

There is already a public registry of the study which can be found here: https://www.hra.nhs.uk/planning-and-improving-research/application-summaries/research-summaries/myms-ally-feasibility-and-acceptability-study-of-ms-intervention/

We hope our rationale is sufficient. Please let me know if we need to discuss this further. Thank you.

Reviewers’ comments:

1. Thank you for noting the contribution of the study in the field of psychological interventions for people with MS.

2. The protocol describes a mixed-method research design with primarily qualitative methods of data collection and analysis. There is no control group and it is not an experimental design therefore the statistical measurements mentioned in this comment do not apply. 

3. The study design is explicitly outlined and provides the context for future research that would follow similar idiographic methods of evaluation of psychological intervention.

4. As this is a protocol for the study, no data has been generated yet. However, data and findings will be available as the project progresses.

5. The article has been further edited.

6. Reviewer 1: 

• We appreciate that CBT has been the prominent psychological intervention for people with MS in the literature. The goal of the current research project is to move away from theory – based interventions and towards a patient – led perspective of intervention development and implementation. We followed the trend in recent literature on clinical therapy processes to move from nomothetic to idiographic approaches of research and intervention (Hofmann & Hayes, 2019). As we have been developing MyMS-Ally group psychological intervention, our aspiration has been to establish processes that correspond to procedures, needs and characteristics of people with MS (Hofmann & Hayes, 2019). Thus, the structure of this intervention has also been based on what people with MS have found significant, helpful and beneficial in their accounts of their experience of psychological interventions they have engaged into (Fragkiadaki et al., 2021). The themes represented the change processes from the people with MS’ perspective as they related them to intervention processes. These themes guided the development of the integrative psychological group intervention MyMS-Ally. However, we acknowledge that as an integrative model we do not suggest an innovative approach therefore we have rephrased it in the article: lines 155, 176, 194. 

References:

Hofmann, S. G., & Hayes, S. C. (2019). The Future of Intervention Science: Process-Based Therapy. Clinical Psychological Science, 7(1), 37–50. https://doi.org/10.1177/2167702618772296

Fragkiadaki, E., Anagnostopoulos, F., & Triliva, S. (2022). The experience of psychological therapies for people with multiple sclerosis: A mixed-methods study towards a patient-centred approach to exploring processes of change. Counselling and Psychotherapy Research, https://doi.org/10.1002/capr.12615

• People with Multiple Sclerosis have access to the internet and are of course able to use devices and gain access to digital resources of support, including attending online interventions.

• The article explicates the objectives of the study presented in this protocol. The detailed outline of the research design and expected outcomes hopefully provides a comprehensive account of what the authors intent to do, which is the scope of this study protocol article.

 Reviewer 2:

Thank you for the encouraging feedback and support of our project.

The reliability information for the scales have been added:

• Satisfaction with Life Scale: p. 17, line 323.

• Hospital Anxiety and Depression Scale: p. 17, line 330.

Thank you for the constructive feedback. We hope the responses and amendments now make the article suitable for publication with Plos One. We are looking forward to hearing back from you.

Kind regards,

Eva Fragkiadaki & co-authors

---

## [Editor Report · Decision Letter 1]

26 Jun 2023

Evaluation of the feasibility and acceptability of an integrative group psychological intervention for people with Multiple Sclerosis: a study protocol

PONE-D-22-34801R1

Dear Dr. Evangelia Fragkiadaki

We’re pleased to inform you that your manuscript has been judged scientifically suitable for publication and will be formally accepted for publication once it meets all outstanding technical requirements.

Kind regards,

GVM Chamath Fernando,

MBBS PgD-FM DipPallMed MCGP MRCGP

Academic Editor

PLOS ONE

<quillbot-extension-portal></quillbot-extension-portal>

---

## [Editor Report · Acceptance letter]

14 Jul 2023

PONE-D-22-34801R1 

Evaluation of the feasibility and acceptability of an integrative group psychological intervention for people with Multiple Sclerosis: a study protocol 

Dear Dr. Fragkiadaki:

I'm pleased to inform you that your manuscript has been deemed suitable for publication in PLOS ONE. Congratulations! Your manuscript is now with our production department. 

Kind regards, 

on behalf of

Dr Gunasekara Vidana Mestrige Chamath Fernando 

Academic Editor

PLOS ONE